# Role of Tyrosine Kinase Syk in Thrombus Stabilisation at High Shear

**DOI:** 10.3390/ijms23010493

**Published:** 2022-01-01

**Authors:** Gina Perrella, Samantha J. Montague, Helena C. Brown, Lourdes Garcia Quintanilla, Alexandre Slater, David Stegner, Mark Thomas, Johan W. M. Heemskerk, Steve P. Watson

**Affiliations:** 1Institute of Cardiovascular Sciences, College of Medical and Dental Sciences, University of Birmingham, Edgbaston, Birmingham B15 2TT, UK; GXP760@student.bham.ac.uk (G.P.); S.J.Montague@bham.ac.uk (S.J.M.); HXB836@student.bham.ac.uk (H.C.B.); Lgquintanilla@yahoo.es (L.G.Q.); a.slater@bham.ac.uk (A.S.); M.R.Thomas@bham.ac.uk (M.T.); 2Department of Biochemistry, CARIM, Maastricht University, 6200 AC Maastricht, The Netherlands; jwmheem722@outlook.com; 3Institute of Experimental Biomedicine I, University Hospital, University of Würzburg, 97080 Würzburg, Germany; stegner@virchow.uni-wuerzburg.de; 4Department Synapse Research Institute, 6214 AC Maastricht, The Netherlands; 5Centre of Membrane Proteins and Receptors (COMPARE), The Universities of Birmingham, Birmingham B15 2TT, UK; 6Centre of Membrane Proteins and Receptors (COMPARE), The Universities of Nottingham, Nottingham NG7 2RD, UK

**Keywords:** disaggregation, platelet, Syk, thrombus, tyrosine kinase

## Abstract

Understanding the pathways involved in the formation and stability of the core and shell regions of a platelet-rich arterial thrombus may result in new ways to treat arterial thrombosis. The distinguishing feature between these two regions is the absence of fibrin in the shell which indicates that in vitro flow-based assays over thrombogenic surfaces, in the absence of coagulation, can be used to resemble this region. In this study, we have investigated the contribution of Syk tyrosine kinase in the stability of platelet aggregates (or thrombi) formed on collagen or atherosclerotic plaque homogenate at arterial shear (1000 s^−1^). We show that post-perfusion of the Syk inhibitor PRT-060318 over preformed thrombi on both surfaces enhances thrombus breakdown and platelet detachment. The resulting loss of thrombus stability led to a reduction in thrombus contractile score which could be detected as early as 3 min after perfusion of the Syk inhibitor. A similar loss of thrombus stability was observed with ticagrelor and indomethacin, inhibitors of platelet adenosine diphosphate (ADP) receptor and thromboxane A_2_ (TxA_2_), respectively, and in the presence of the Src inhibitor, dasatinib. In contrast, the Btk inhibitor, ibrutinib, causes only a minor decrease in thrombus contractile score. Weak thrombus breakdown is also seen with the blocking GPVI nanobody, Nb21, which indicates, at best, a minor contribution of collagen to the stability of the platelet aggregate. These results show that Syk regulates thrombus stability in the absence of fibrin in human platelets under flow and provide evidence that this involves pathways additional to activation of GPVI by collagen.

## 1. Introduction

Platelet-rich thrombi formed in vivo in mice have been shown to be composed of a core and shell region in both the arterial and venous microcirculation [1]. In the core region, platelets and fibrin are densely packed, and permeability is heavily restricted [2], necessitating the use of a fibrinolytic to dissolve the platelet aggregate [3,4]. The outer, more permeable, shell region however, consists of weakly activated and loosely packed platelets and does not contain fibrin [2], suggesting that a different strategy is needed to promote disruption of this region and thus help to prevent the build-up of an occlusive thrombus. 

Glycoprotein VI (GPVI) is a platelet immunoglobulin receptor which is known as a receptor for collagen [5], but in recent years has also been shown to be a receptor for fibrin and fibrinogen, among other predominantly charged ligands [6]. In vitro flow studies have shown that GPVI contributes to the stability of newly formed platelet aggregates on collagen at high shear through use of the blocking anti-GPVI Fab ACT017, which has more recently been named glenzocimab [4]. Since the activation of GPVI by fibrinogen is also dependent on integrin αIIbβ3, with the interplay of the two receptors driving platelet adhesion and activation [7]; this suggests that blocking signalling pathways common to both receptors may have a greater antithrombotic effect than blocking GPVI alone. 

The tyrosine kinase Syk plays a critical role in signalling by integrin αIIbβ3 and GPVI [8,9] and an inhibitor of Syk, fostamatinib, is clinically used for the treatment of patients with refractory immune thrombocytopenia (ITP) without increasing the risk of bleeding despite the marked reduction in platelet count in this patient group [10]. Moreover, a retrospective analysis provides evidence of a notably low incidence of thrombotic events in patients treated with the Syk inhibitor [10]. Thus, Syk inhibitors represent a new class of antiplatelet agent with reduced bleeding risk compared to current drugs. The involvement of Syk in thrombus formation and thrombus growth and stabilisation suggests that inhibitors will be effective against multiple stages in thrombus formation.

In the present study, we have investigated the role of Syk on the stability of preformed thrombi formed on collagen or human atherosclerotic plaque homogenate at arterial shear. Furthermore, we have compared the effect of Syk inhibition to that of the feedback agonists, adenosine diphosphate (ADP) and thromboxane A_2_ (TxA_2_), and of the tyrosine kinases Src and Btk. We have also investigated the effect of a recently described nanobody, Nb21, to GPVI that blocks platelet activation by collagen [11]. The results show that inhibition of Syk promotes thrombus breakdown on collagen and plaque homogenate thereby demonstrating a critical role for the kinase in the stability of this region.

## 2. Materials and Methods

### 2.1. Materials

The complete list of reagents is described in Appendix A. The anti-GPVI nanobody 21 (Nb21) was generated as described [11]. The human plaque homogenate was obtained from 10 patients with symptomatic carotid artery stenosis undergoing carotid endarterectomy, then homogenised and pooled as described [12]. All experiments involving human subjects were performed in accordance with the declaration of Helsinki and Good Clinical Practice and approved by NHS research and ethics committees (North West—Haydock Research Ethics Committee 20/NW/0001 and West Midlands—South Birmingham Research Ethics Committee 18/WM/0386). Use of blood from healthy volunteers was approved by the University of Birmingham Ethics Review (ERN_11-0175AP10). An informed consent was obtained from the healthy volunteers and patients to participate in the study.

### 2.2. Blood Withdrawal and Platelet Preparation

Blood was taken into 4% sodium citrate from consenting healthy volunteers, who had not taken anti-platelet agents in the previous ten days. Washed human platelets were obtained by centrifugation of platelet-rich plasma (PRP) in the presence of prostacyclin (0.56 µM) and resuspended in modified Tyrode’s buffer (134 mM NaCl, 0.34 mM Na_2_HPO_4_, 2.9 mM KCl, 12 mM NaHCO_3_, 20 mM Hepes, 5 mM glucose, 1 mM MgCl_2_, pH 7.3) as described [13]. The isolated platelets were used at a cell density of 5 × 10^8^/mL for protein phosphorylation studies and of 2 × 10^8^/mL for aggregometry.

### 2.3. Platelet Aggregation

Changes in light transmission were recorded at 37 °C with stirring (1200 rpm) in an optical aggregometer (PAP-8E, Milan, Italy). Washed platelets, in presence or absence of eptifibatide (9 µM), were warmed for 2 min prior to activation by Horm collagen (30 µg/mL), thrombin (1 U/mL), fibrinogen (200 µg/mL) or TRAP (10 µM).

### 2.4. Western Blotting

Platelet activation was terminated with a sodium dodecyl sulphate (SDS) reducing sample buffer. Whole cell lysates (WCL) were prepared and immunoblotted as previously described [14]. Syk and PLCγ2 were immunoprecipitated as previously described [14]. 

### 2.5. Flow Adhesion

Glass coverslips were coated overnight at 4 °C with Horm collagen (50 µg/mL) or plaque homogenate (0.5 mg/mL) before blocking with 10% BSA for 30 min in Hepes buffer (10 mM Hepes, 136 mM NaCl, 2.7 mM KCl, 2 mM MgCl_2_, 5.5 mM glucose, and 0.1% BSA), pH 7.45 [7]. Citrated whole blood, recalcified with CaCl_2_ (7.5 mM) and MgCl_2_ (3.75 mM), in the presence of PPACK (40 µM), was perfused for 7 min at 1000 s^−1^ [15]. Aggregates were successively post-perfused with Hepes buffer with vehicle (DMSO, 0.01%) or an inhibitor of Syk (PRT-060318, 10 µM), Src (dasatinib, 10 µM), Btk (ibrutinib, 7 µM), GPVI (Nb21, 500 nM), P2Y_12_ (ticagrelor, 10 µM), or cyclooxygenase (indomethacin, 10 µM), for 10 min at 1000 s^−1^. Inhibitors were added to the Hepes buffer pH 7.45, supplemented with 0.1% BSA, 0.1% glucose, 2 mM CaCl_2_ and 1 U/mL heparin. Brightfield images of a same field of view, which included multiple aggregates, were taken every 60 s for 10 min. Aggregate morphology, contraction and multilayer scores were assessed blind by visual inspection compared to a standard set of representative images [16]. This scoring system distinguishes single adhesive platelets from those organized in aggregates (morphological score) and monitors the tightness (contraction score) and size of the aggregates (multilayer score). Where stated experiments were performed at 37 °C.

### 2.6. Epifluorescence Microscopy

Recalcified blood, treated with PPACK and labelled with PE anti-GPVI mAb, APC anti-CD41a mAb and FITC anti-fibrinogen mAb was flowed for 7 min at arterial shear (1000 s^−1^) over coverslips coated with Horm collagen (50 µg/mL) or plaque homogenate (0.5 mg/mL). The formed aggregates were post-perfused with vehicle (DMSO, 0.01%) or PRT-060318 (10 µM) for 3 min at arterial shear (1000 s^−1^), and then fixed in formalin for 15 min, before washing in PBS and mounting in between glass slides. Seven images per fields of view were taken using an epifluorescence microscope. Image analysis was performed by measuring fluorescence intensity in the open source software Fiji [17].

### 2.7. Statistical Analysis

Data are presented as mean ± s.e.m. unless stated otherwise, with statistical significance taken at *p* < 0.05 (one-tailed Student’s paired *t*-test). Statistical analyses were performed using GraphPad Prism 7 (GraphPad Software Inc. La Jolla, CA, USA).

## 3. Results

### 3.1. Thrombin Stimulates Sustained Phosphorylation of Syk via Integrin αIIbβ3 during Platelet Aggregation

The outer shell of an arterial thrombus is composed of aggregated platelets held together by the interaction of fibrinogen with integrin αIIbβ3, with activation of the integrin mediated by the autocrine feedback mediators, ADP and TxA_2_ [18,19]. It has been proposed that the two mediators diffuse from the highly activated platelets in the thrombus core region to support platelet aggregation in the shell [20,21]. In this model, it is unclear how the activation of the integrin is maintained over time as the levels of the two mediators decline due to exhaustion of intracellular stores and reduced activation of phospholipase A_2_. One potential explanation is that integrin activation is maintained by a positive feedback pathway mediated through the activation of αIIbβ3 and GPVI by fibrinogen [7]. If this is the case, the activation of Syk during platelet aggregation should be sustained.

To investigate this, we activated platelets in suspension by thrombin which signals through the G_q_ protein-coupled receptors, PAR-1 and PAR-4 [22]. Thrombin was selected for the experiments because it is the most powerful G_q_ protein-coupled receptor ligand in platelets. We performed the experiments in the absence or presence of the αIIbβ3 antagonist, eptifibatide, to establish the role of the integrin in mediating Syk activation. 

Thrombin evoked sustained tyrosine phosphorylation of Syk over 150–3000 s (as concluded from 4G10 staining after Western blotting), which was inhibited by eptifibatide at all time points (Figure 1(Ai–iv)). In contrast, collagen-induced tyrosine phosphorylation of Syk, was reduced but not blocked by eptifibatide (Figure 1(Ai,ii)). Eptifibatide also blocked the increase in tyrosine phosphorylation of LAT and PLCγ2 by thrombin but not by collagen (Figure 1(Bi–iv)). 

To rule out a possible role of fibrin generation by thrombin in these experiments, the studies were repeated with the PAR-1 peptide agonist TRAP (Figure 1(Ci)). TRAP also induced a sustained phosphorylation of Syk, LAT and PLCγ2 (Figure 1(Cii–iv)), whereas fibrinogen had no effect, as expected. 

These data show a sustained activation of Syk downstream of integrin αIIbβ3 raising the possibility that this contributes to the maintenance of integrin activation during platelet aggregation. 

### 3.2. Inhibition of Syk on Collagen and Plaque Homogenate Causes Thrombus Shell Instability

Experiments were designed to investigate the contribution of Syk to GPVI-dependent thrombus stability. In these experiments, recalcified blood was perfused for 7 min at arterial shear (1000 s^−1^) over Horm collagen or human plaque material. Both ligands stimulate platelet activation through GPVI [5,12]. The thrombin inhibitor, PPACK, was used to prevent fibrin formation. 

Aggregates were post-perfused for 10 min with vehicle (DMSO) or the Syk inhibitor, PRT-060318, at a concentration (10 µM) that has been shown to be effective in whole blood [23,24].

Heparin was added to the post-perfusion buffer to prevent residual formation of fibrin. Thrombus disaggregation was monitored by taking brightfield images at 60 s intervals. 

Aggregates formed after 7 min of blood perfusion on both collagen and plaque material were tightly contracted and consisted of multiple platelet layers as illustrated in Figure 2A,B. Post-perfusion with vehicle for 10 min had a minor effect on the architecture, as reflected by the small reduction in morphological and contraction scores (Figure 2(Ci,ii,Di,ii)). The aggregates were still composed of multiple layers of platelets as shown by the multilayer score (Figure 2(Ciii,Diii)). In contrast, perfusion with PRT-060318 on both surfaces promoted aggregate breakdown which could be clearly seen by eye (Figure 2A,B) and was confirmed by measurement of morphology (Figure 2(Ci,Di)). The increased breakdown could be detected within 3 min as shown by the reduction in the contraction score at this time (Figure 2(Cii,Dii)). The contraction score had return to base-line by 6–10 min. The loss of stability led to a slow detachment of platelets eventually forming a monolayer on the surface (Figure 2(Ciii,Diii)). 

To further explore the effect of Syk inhibition on thrombus characteristics, recalcified blood was triple labelled with PE-anti-GPVI mAb, APC-anti-CD41a mAb and FITC-anti-fibrinogen prior to perfusion over collagen and plaque, and then post-perfused for 3 min in the presence of buffer or PRT-060318 (Figure 3(Ai,ii)). PPACK and heparin were included as above to prevent thrombin formation. At this early time, perfusion with the Syk inhibitor caused a significant impairment in morphological, contraction and multilayer scores (Figure 3B). Fluorescence images of the aggregates indicated a reduced labelling of fibrinogen and GPVI staining in the presence of PRT-060318 although this did not reach significance (Figure 3C). Imaging was not performed at later timepoints due to the reduction in intensity through loss of platelets. 

Taken together, the above results show that post-inhibition of Syk promotes disaggregation of the preformed thrombi on surfaces of collagen or plaque which eventually leads to the formation of a platelet monolayer.

### 3.3. Inhibition of Syk Promotes Thrombus Breakdown at 37 °C

We next asked if the contribution of Syk to thrombus shell architecture is also seen at the body temperature of 37 °C. The thrombi formed at 37 °C were larger and more tightly packed than those formed at room temperature (Figure 4A). In the presence of PRT-060318, the aggregates could be seen to loosen and to spread out from approximately 2 min of perfusion with the Syk inhibitor relative to vehicle (Figure 4(Bi,ii)). However, the majority of platelets was retained in the aggregate for up to 5 min and only at later times could they be seen to detach resulting in a significant decrease in the multilayer score (Figure 4(Bi,ii)). 

This shows that Syk contributes to thrombus stability at both room temperature and at 37 °C temperature despite the increased stability of the aggregates at the higher temperature.

### 3.4. Inhibition of Src and Secondary Agonists but Not Btk Promotes Thrombus Breakdown

We then extended the studies at 37 °C to investigate the role of Src and Btk tyrosine kinases in aggregate stability, alongside the secondary mediators ADP and TxA_2_. Src kinases lie both upstream and downstream of Syk in the GPVI signalling cascade, and Btk lies downstream of both kinases [25].

As observed with the Syk inhibitor PRT-060318, the Src inhibitor dasatinib (10 µM) induced an increase in aggregate breakdown at 2 min of post-perfusion (Figure 4(Ci,ii)), with a marked loss of aggregate architecture after 6 min of post-perfusion similar to that observed in the presence of PRT-060318 (Figure 4B,C). In contrast, the Btk inhibitor ibrutinib (7 µM), at a concentration that is effective in plasma [26] caused only a mild decrease in the contractile score and no apparent change in morphology (Figure 4D). The much weaker effect may be due to redundancy with the Tec family kinase, Tec, or the later role of Btk in the GPVI signalling cascade. 

The dependency of thrombus stability on the secondary mediators, ADP and TxA_2_, was investigated using the P2Y_12_ receptor antagonist ticagrelor (10 µM) and cyclooxygenase inhibitor indomethacin (10 µM), respectively. The two inhibitors had a similar effect on thrombus stability to that of PRT-060318 and dasatinib (Figure 4(Bii,Cii)). Moreover, the combination of indomethacin, ticagrelor and PRT-060318 caused only a slight increase in disaggregation suggesting that the two mediators work in concert with Syk to promote aggregation (Figure 4(Bii,Cii)). This could reflect either synergy of intracellular signals or a role of Syk in stimulating release of ADP and TxA_2_.

Together, the results show a critical role for Syk and Src kinases alongside the two secondary mediators ADP and TxA_2_ in supporting aggregate stability on collagen.

### 3.5. Blocking of GPVI with Nanobody 21 Causes Only a Minor Thrombus Breakdown on Collagen

The nanobody Nb21 was raised against human GPVI and shown to block platelet activation by collagen as measured by light transmission aggregometry and flow adhesion over collagen at arterial shear [11]. Post-perfusion with Nb21 however caused only a small impairment in thrombus stability over collagen at longer times of flow (8 min; Figure 5B), suggesting that loss of stability in the presence of Syk and Src inhibitors is largely independent of blockade of GPVI activation by collagen.

## 4. Discussion

Platelets in the shell region of aggregates formed in vivo are held together by the binding of fibrinogen to the major platelet integrin αIIbβ3, with the diffusion of secondary mediators from the thrombus core region mediating inside-out activation of the integrin and binding to fibrinogen. The secretion of the two mediators diminishes over time leading to the question as to whether alternative pathways exist to support integrin activation in the shell. 

In the present study, we demonstrate a critical role for Src and Syk tyrosine kinases in supporting the stability of preformed shell-type thrombi on collagenous surfaces in the absence of fibrin formation, and show that this is mediated in concert with the signals from ADP and TxA_2_. We propose that activation of Src and Syk kinases is mediated through outside-in signalling by αIIbβ3 and GPVI through their interaction with fibrinogen leading to a feedback pathway that reinforces integrin activation in combination with ADP and TxA_2_. This is supported by the observation in the present study that activation by the G protein-coupled receptor ligands thrombin or TRAP induces weak but sustained activation of Syk through an αIIbβ3-dependent pathway. Moreover, the initial activation of GPVI by collagen appears to play little role in thrombus stability as shown using a nanobody, Nb21, that blocks activation of GPVI by matrix proteins. The effect on thrombus stability is also independent of adhesion as a single layer of platelets is retained in the presence of the inhibitors. The results emphasise the critical interplay of tyrosine kinase-based and G protein-coupled receptors in supporting the stability of preformed thrombi under containing a shell-type of aggregated platelets under higher shear. 

An important consideration is the extent to which results from the in vitro flow model can be extrapolated to formation of the shell region in vivo. The shell region is distinguished from the core by the absence of fibrin, and is composed of loosely packed, aggregated platelets which, at the extremes of the shell, retain their discoid shape suggesting that they are less active. It is this region that has been modelled in the flow experiments in this study due to the presence of PPACK which prevents thrombin and fibrin formation. However, in the in vitro flow model, the aggregate is formed on a layer of collagen or plaque material and not on a fibrin-rich aggregate.

A critical role for TxA_2_ and the ADP P2Y_12_ receptor in the formation of the shell in vivo has been reported [18] and has been replicated here in the flow model. Under arterial flow in vitro and in the absence of fibrin, inhibition of Syk has been shown to impair thrombus formation on collagen [23] and on fibrinogen, [7] suggesting a critical inter-play of integrin αIIbβ3 and GPVI in platelet activation. Further, continuous signalling through ADP has been shown to prevent platelet disaggregation in vitro by supporting binding of integrin αIIbβ3 to fibrinogen [27]. There exists a wealth of reports of synergy between G protein-coupled and tyrosine kinase-linked receptors in mediating platelet activation suggest that this is also likely to be replicated in vivo. However, fibrinogen does not activate GPVI in mouse platelets [28], and the important role of Syk in thrombus formation [29,30] makes it difficult to investigate the role of Syk in thrombus stability in vivo in mice.

The present results delve deeper into the recent observations of Ahmed et al. [4] that the GPVI antagonist, ACT017, destabilises preformed thrombi through a pathway that is dependent on downstream signalling. The inability of Nb21 to mimic the effect of Syk inhibition is in contrast to the result with ACT017 and can be explained by the lack of effect of Nb21 on fibrinogen-dependent activation of GPVI (unpublished observation), in contrast to that of ACT017 [4]. This provides indirect evidence that the two inhibitors bind to distinct sites on GPVI.

GPVI has long been recognised to be a promising anti-thrombotic target for many years, [31] but as of today, ACT017 (glenzocimab) is the only anti-GPVI blocker currently in clinical trial, where it is being used in combination with best treatment for acute stroke [32]. Inhibitors of αIIbβ3 are widely used against acute coronary syndrome but are not suitable for long-term therapy due to the risk of excessive bleeding [33]. Src, Syk and Btk inhibitors are currently available in clinic for the treatment of patients with solid tumour, ITP and haematological malignancies, and have been shown to be well tolerated [34]. Both GPVI and αIIbβ3 activate Src and Syk kinase suggesting that blocking either kinase may significantly impair thrombus formation while overcoming limitations concerning inhibitors of both receptors. Moreover, neither the Src or Syk inhibitors alter the response to G protein-coupled receptors signals which are critical for haemostasis. 

The present results show similar effects with inhibitors of Syk and Src kinases but not with ibrutinib which blocks Btk. The latter may be due to redundancy with Tec or its position further down the GPVI signalling cascade. Src kinases are ubiquitous and as yet selective inhibitors have not been identified, and current inhibitors have a wide spectrum of off-target effects. On the other hand, Syk is restricted to haematopoietic cells. Further, the only inhibitor in the clinic, fostamatinib, has been shown to be well tolerated even in ITP patients with low platelet counts, [35] and there is evidence of a reduction in thrombus formation in patients on prolonged treatment with fostamatinib [10].

In conclusion, the present results show a critical role for Syk in supporting thrombus stability under arterial shear and provide evidence that this is independent of activation of GPVI by collagen. The findings further emphasise Syk as a target for development of a new class of antiplatelet agent that may prevent formation of occlusive thrombosis by both inhibiting thrombus formation and by promoting thrombus breakdown in areas of weak platelet activation. 

## Figures and Tables

**Figure 1 ijms-23-00493-f001:**
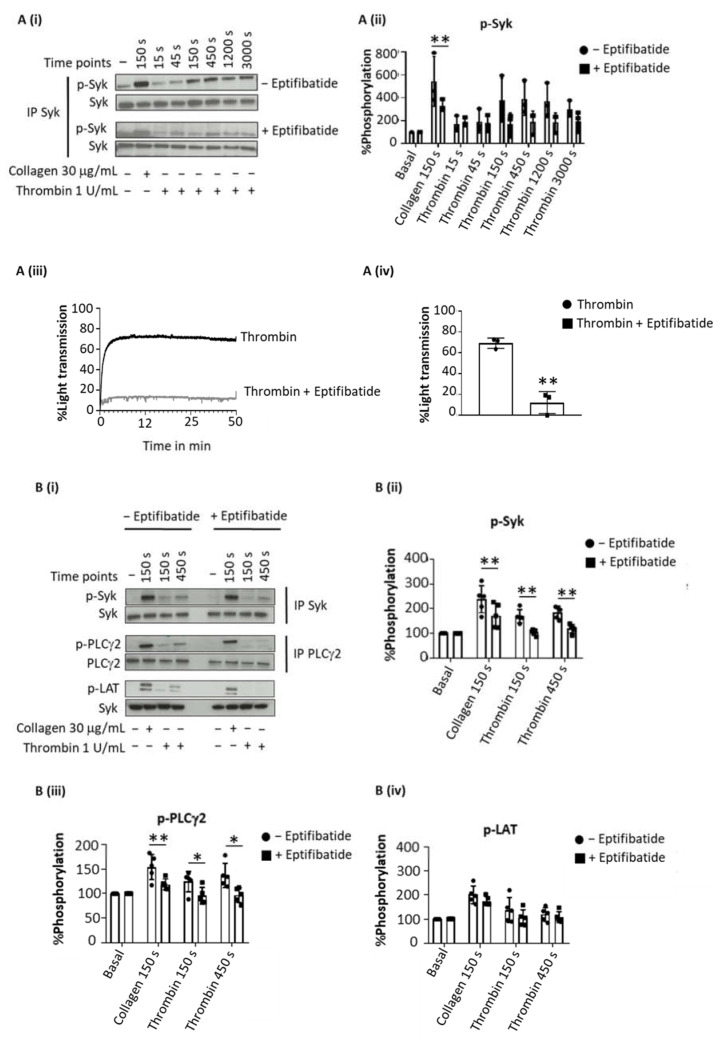
Sustained Syk phosphorylation in thrombin-stimulated platelet is integrin-dependent. Washed platelets at 5 × 10^8^/mL were stimulated with Horm collagen (30 µg/mL), thrombin (1 U/mL), TRAP (10 µM) or fibrinogen (200 µg/mL) in the presence or absence of eptifibatide (9 µM). Activation was stopped at stated time after addition of the agonist. (**A**) Representative blot of Syk tyrosine phosphorylation from 6 donors. (**Ai**) Whole cell lysates were immunoprecipitated with anti-Syk mAb and probed with anti-phospho-tyrosine mAb 4G10; (**Aii**) Percentage of Syk tyrosine phosphorylation, based on Western blots, from 6 donors (mean ± s.d.); (**Aiii**) Aggregation trace of platelets stimulated with thrombin (1 U/mL), representative of 3 donors, and (**Aiv**) graph showing aggregation as % light transmission (mean ± s.d.) after 50 min of agonist stimulation. (**B**) Representative Western blot (**Bi**), and calculated percentage of Syk (**Bii**), PLCγ2 (**Biii**) and LAT (**Biv**) phosphorylation in response to thrombin (Thromb) from 5 donors (mean ± s.d.). Horm collagen (Col) was used as positive control. (**C**) Representative blot (**Ci**); mean ± s.d. of Syk (**Cii**), PLCγ2 (**Ciii**) and LAT (**Civ**) phosphorylation in response to collagen (Col), TRAP or fibrinogen (Fgn) from 3 donors. * *p* < 0.05, ** *p* < 0.005, one-tailed paired Student’s *t*-test. Where stated, proteins were immunoprecipitated (IP).

**Figure 2 ijms-23-00493-f002:**
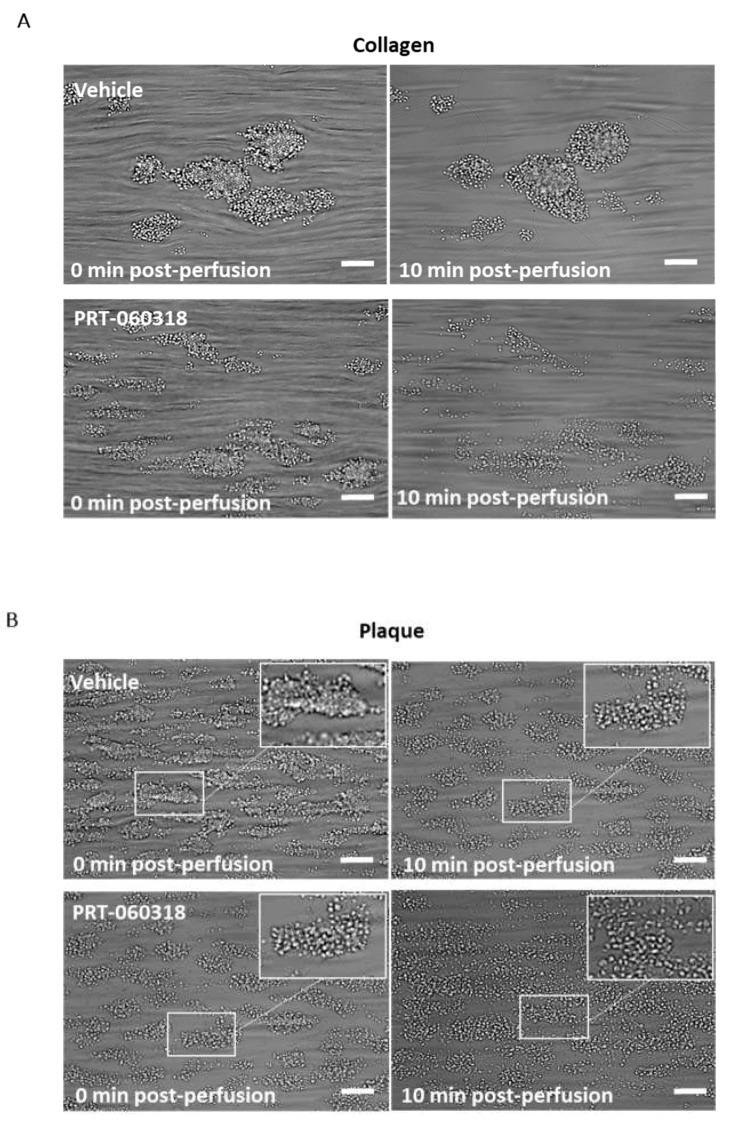
Inhibition of Syk causes loss of contraction and platelet adhesion on collagen and plaque material. Recalcified blood from 6 donors was perfused for 7 min on Horm collagen and plaque at room temperature at a shear rate of 1000 s^−1^. (**A**,**B**) Representative images of aggregates formed on collagen (**A**) and plaque (**B**), taken during post-perfusion at times 0 and 10 min with vehicle or with PRT-060318 (*n* = 6). Scale bar = 100 µm. (**C**,**D**) Disaggregation on collagen (**C**) or on plaque (**D**) was monitored by taking brightfield images every 60 sec and measured using morphological (**Ci**,**Di**), contraction (**Cii**,**Dii**) and multilayer scores (**Ciii**,**Diii**) [16]. Aggregates were post-perfused with rinse buffer ± vehicle (DMSO) or the Syk inhibitor PRT-060318 (10 µM). Data are shown as mean ± s.e.m. * *p* < 0.05, ** *p* < 0.005.

**Figure 3 ijms-23-00493-f003:**
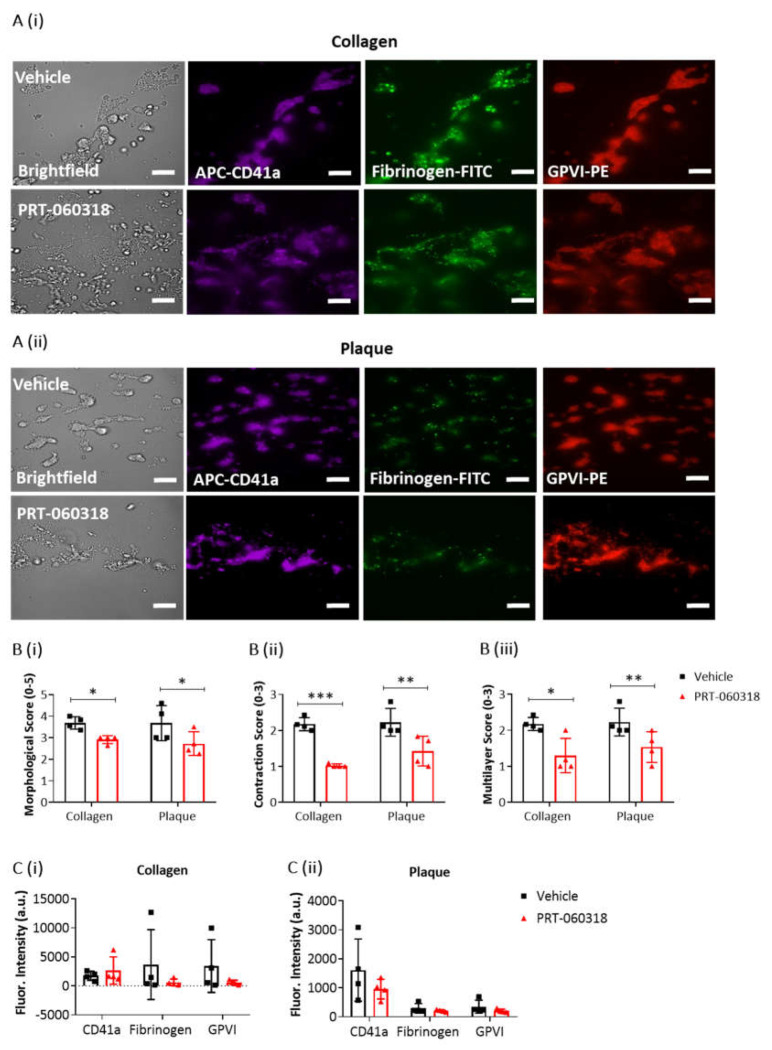
Platelet activation imaging of thrombi postperfused under condition of Syk inhibition. Recalcified blood samples labelled with APC-αCD41a mAb (purple), FITC-fibrinogen (green) and PE-αGPVI mAb (red) was perfused for 7 min over immobilised collagen or plaque material at room temperature and arterial shear rate (1000 s^−1^). Post-perfusion for 3 min was with rinse buffer containing vehicle (DMSO) or Syk inhibitor PRT-060318 (10 µM). (**A**) Representative microscopic images, showing remaining aggregates on collagen (**Ai**) or plaque (**Aii**) after 3 min of post-perfusion with vehicle or PRT-060318 (*n* = 4 donors). (**B**) Graphs showing scores of thrombus morphology (**Bi**), thrombus contraction (**Bii**) and thrombus multilayer (**Biii**) [16]. (**C**) Graphs of fluorescence intensity (arbitrary units, a.u.) of platelet aggregates on collagen (**Ci**) or plaque material (**Cii**) fixed after 3 min of post-perfusion. Scale bar = 50 µm. Data are shown as mean ± s.d., * *p* < 0.05, ** *p* < 0.005, *** *p* < 0.0005, one-tailed Student’s paired *t*-test.

**Figure 4 ijms-23-00493-f004:**
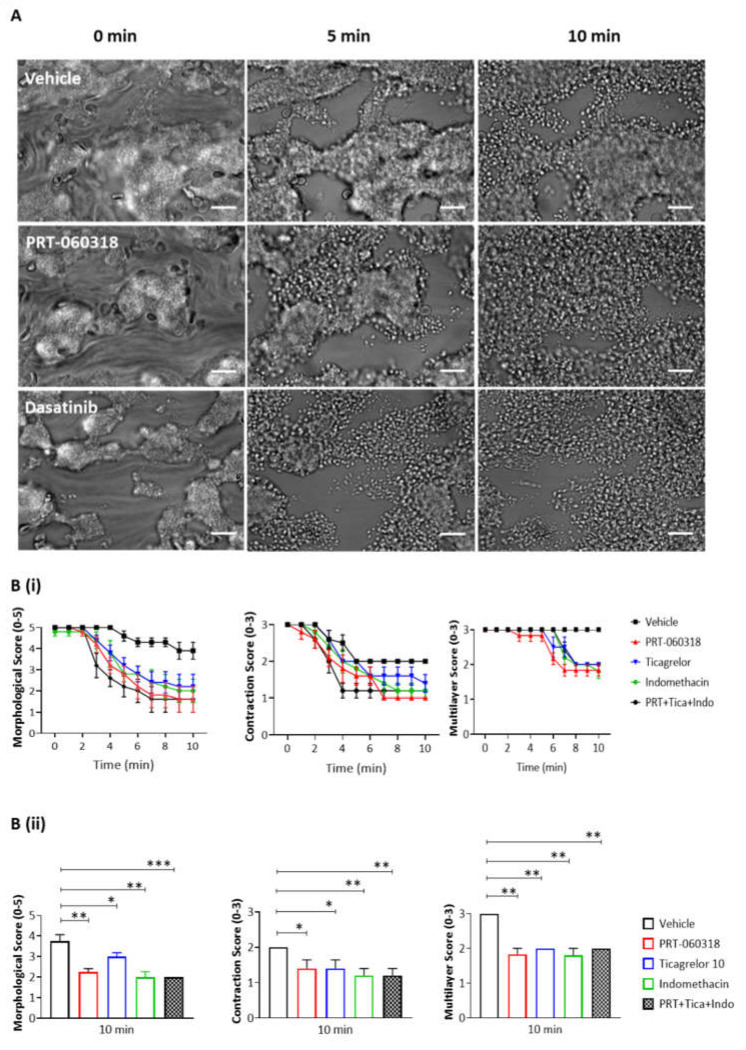
Inhibition of Syk causes the same extent of disaggregation observed with antagonists of adenosine diphosphate (ADP) and thromboxane A_2_ (TxA_2_). Recalcified blood was perfused for 7 min on Horm collagen at 37 °C at arterial shear (1000 s^−1^), and post-perfused for 10 min with rinse buffer with vehicle (DMSO) or inhibitor of Syk, Src or Btk (PRT-060318 10 µM, dasatinib 10 µM or ibrutinib 7 µM, respectively). (**A**) Representative images of aggregates at 0, 5, and 10 min of post-perfusion with vehicle, PRT-060318 (*n* = 5) or dasatinib (*n* = 6). (**B**–**D**) Graphs showing extent of disaggregation, as measured from scores of thrombus morphology, contraction and multilayer,^16^ at every 60 s (**Bi**,**Ci**,**D**) or at 10 min (**Bii**,**Cii**) of post-perfusion with either vehicle, PRT-060318 (PRT) (**Bi**,**ii**), dasatinib (**Ci**,**ii**) or ibrutinib (**D**), alone or combined with indomethacin (Indo, 10 µM) and ticagrelor (Tica, 10 µM). Scale bar = 50 µm. Data are shown as mean ± s.e.m. * *p* < 0.05, ** *p* < 0.005, *** *p* < 0.0005, one-tailed paired Student’s *t*-test.

**Figure 5 ijms-23-00493-f005:**
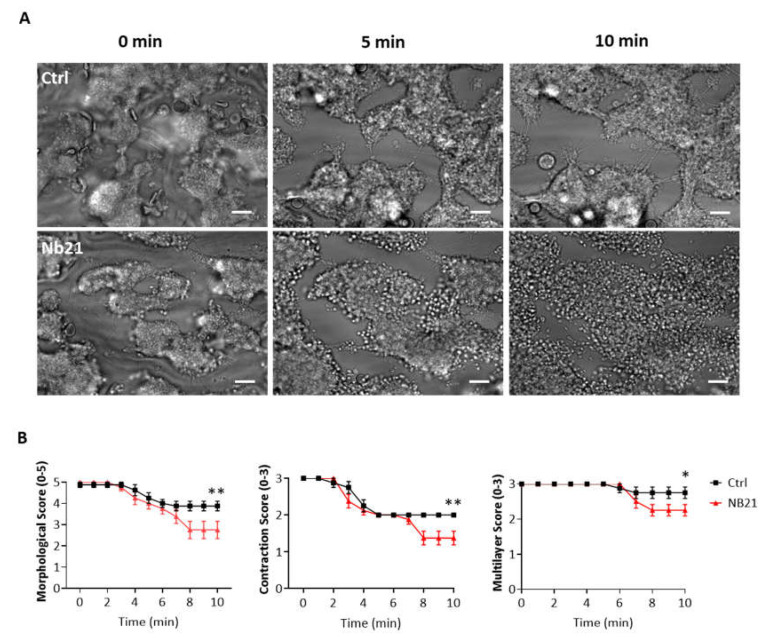
Blockage of GPVI with the anti-GPVI nanobody 21 (Nb21) causes minor thrombus instability. Recalcified blood was perfused for 7 min on Horm collagen at 37 °C at arterial shear (1000 s^−1^), and successively post-perfused for 10 min with rinse buffer and blocking anti-GPVI nanobody Nb21 (500 nM). (**A**) Representative images after perfusion of blood from 6 donors, taken during post-perfusion at time 0, 5 and 10 min. (**B**) Graphs showing extent of disaggregation, monitored from recorded brightfield images every min, and measured as morphological, contraction and multilayer scores [16]. Scale bar = 50 µm. Data are shown as mean ± s.e.m. * *p* < 0.05, ** *p* < 0.005 one-tailed Student’s paired *t*-test.

## Data Availability

Not applicable.

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
