# Peer review of "Role of Tyrosine Kinase Syk in Thrombus Stabilisation at High Shear"

_ijms, 2022, doi:10.3390/ijms23010493_

Round 1

Reviewer 1 Report

Finding new effective and safe antiplatelet agents, preventing thrombosis with minor bleeding risk is an important goal for cardiovascular health.

In this study, Perrella et al. , by in vitro assays, show the important role of Syk and src  in the stability of the shell region of thrombus. Inhibition of this kinases will therefore causes thrombus instability and can, therefore, be seen as potential antiplatelet agents. Syk is preferable to src, as its expression, in contrast to that of src, but s restricted to hematopoietic cells.

The authors show that the role of Syk in thrombus stability rely in its sustained activation downstream of aIIbb3 signaling and it is independent of GPVI signalling.      

Overall the study is well performed, results are clear and the manuscript is easy to read.

I have no major issues on the study and the paper. 

Minor comments:

  • Figure legends should be reviewed to assess clarity and correspondence  with text..  For example, in text and Fig 4 legend  there is no or unclear mention to panel D ( Di-ii?); In Fig 5 Legend, I guess “perfusion” should be  “perfusion”
  • In Discussion, please soften the claim that “Both GPVI and αIIbβ3 activate Src and Syk kinase suggesting that blocking either kinase may exert a stronger effect on thrombus formation than blocking either receptor alone ..”In my opinion current inhibition with anti- αIIbβ3  agents currently in the clinic is so strong that  would mask  the effect of any other drug.

Author Response

Finding new effective and safe antiplatelet agents, preventing thrombosis with minor bleeding risk is an important goal for cardiovascular health.

In this study, Perrella et al. , by in vitro assays, show the important role of Syk and src  in the stability of the shell region of thrombus. Inhibition of this kinases will therefore cause thrombus instability and can, therefore, be seen as potential antiplatelet agents. Syk is preferable to src, as its expression, in contrast to that of src, but s restricted to hematopoietic cells.

The authors show that the role of Syk in thrombus stability rely in its sustained activation downstream of aIIbb3 signaling and it is independent of GPVI signalling.     

Overall the study is well performed, results are clear and the manuscript is easy to read.

I have no major issues on the study and the paper.

We thank the reviewer for the positive comment on our work. 

Minor comments:

  1. Figure legends should be reviewed to assess clarity and correspondence with text. For example, in text and Fig 4 legend there is no or unclear mention to panel D (Di-ii?); In Fig 5 Legend, I guess “perfusion” should be “perfusion”.

We have checked and edited all the figure legends.

  1. In Discussion, please soften the claim that “Both GPVI and αIIbβ3 activate Src and Syk kinase suggesting that blocking either kinase may exert a stronger effect on thrombus formation than blocking either receptor alone..”In my opinion current inhibition with anti- αIIbβ3 agents currently in the clinic is so strong that  would mask  the effect of any other drug.

Thank you for this important comment. We have edited the sentence in the discussion accordingly.

Reviewer 2 Report

Perrella and colleagues have presented very interesting findings on the role of Syk in platelet aggregate stability with important implications for arterial thrombosis. The approach and observations are exciting, novel, and could have a valuable impact on the future development of next generation antithrombotics that have reduced bleeding risks. I have just a few comments/questions:

1) An inducible Syk knockout mouse currently exists. Does this mouse show resistance, reduction, or enhance degradation of arterial thrombi? If this has not been done, do the authors plan to undertake such experiments?

2) Alternatively, have the authors attempted to recapitulate these in vitro findings in vivo using the reported inhibitors? Either in WT or atherosclerotic prone rodents?

3) In these experiments, Src and Syk inhibitors were tested on pre-formed thrombi, however was there an impact on thrombus formation given the contribution to platelet aggregation? 

4) Figure legends were not included which made interpretation of data challenging.

Author Response

Perrella and colleagues have presented very interesting findings on the role of Syk in platelet aggregate stability with important implications for arterial thrombosis. The approach and observations are exciting, novel, and could have a valuable impact on the future development of next generation antithrombotics that have reduced bleeding risks.

We are grateful for the positive comments on our work. 

Comments:

  1. An inducible Syk knockout mouse currently exists. Does this mouse show resistance, reduction, or enhance degradation of arterial thrombi? If this has not been done, do the authors plan to undertake such experiments?

We do not have access to this mouse at the present. The results on thrombus stability would be difficult to interpret due to the role of Syk in thrombus formation. A previous work from van Eeuwijk et al. showed a reduced thrombus formation in mice lacking Syk and a similar result was obtained by Andre et al. when looking at thrombus formation ex vivo. In addition to this limitation, fibrinogen does not activate GPVI in mouse platelets. We revised the discussion accordingly.

  1. Alternatively, have the authors attempted to recapitulate these in vitro findings in vivo using the reported inhibitors? Either in WT or atherosclerotic prone rodents?

We agree that this is an interesting experiment, but it has the same limitation as above. 

  1. In these experiments, Src and Syk inhibitors were tested on pre-formed thrombi, however was there an impact on thrombus formation given the contribution to platelet aggregation?

Inhibition of Syk has been reported to block thrombus formation – this is now mentioned in the revised manuscript in the Discussion along with a supporting reference.

  1. Figure legends were not included which made interpretation of data challenging.

We are not sure why this is the case as Referee No. 1 was able to access them.